# OpenReview forum: "LatentBreak: Jailbreaking Large Language Models through Latent Space Feedback"
_ICLR.cc/2026/Conference — ICLR 2026 Conference Withdrawn Submission_

### Official Review · Reviewer_oY46 · 2025-10-24

**Soundness:** 2
**Presentation:** 3
**Contribution:** 2
**Rating:** 4
**Confidence:** 3

**Summary:**

This paper introduces LatentBreak, a white-box jailbreak attack that generates natural, low-perplexity adversarial prompts through latent-space feedback. Instead of appending adversarial suffixes or long templates, the method iteratively substitutes words in the harmful prompt with semantically similar alternatives, guided by a latent-space distance objective that moves the prompt’s representation toward a “harmless” centroid. Experiments across various LLMs and multiple defenses demonstrate the effectiveness of LatentBreak, while maintaining shorter prompts and evading perplexity-based filters.

**Strengths:**

- The paper is well-written, clearly structured, and easy to follow.

- The motivation that evades perplexity-based detectors by generating low-perplexity jailbreak prompts is intuitively sound and practically relevant.

**Weaknesses:**

- Limited novelty beyond representation shifting:
The central idea—shifting the prompt’s latent representation toward the “harmless” region—is conceptually similar to prior work on representation-space analysis of jailbreaks (e.g., Towards Understanding Jailbreak Attacks in LLMs: A Representation Space Analysis, Li et al., 2025; Refusal in LMs Is Mediated by a Single Direction, Arditi et al., 2024). Compared to these studies, LatentBreak mainly turns the representation insight into a heuristic substitution algorithm. While effective, this feels incremental rather than a fundamental advance.

- Relation to prior work:
Compared with recent representation-based jailbreak studies, this paper provides less interpretability or insight into the safety mechanism itself. It would benefit from a more analytical discussion connecting its latent-shift behavior to known refusal vectors or alignment subspaces.

- Baseline selection is dated:
The paper mainly compares against earlier 2023–2024 jailbreaks (GCG, GBDA, SAA, AutoDAN, PAIR, TAP). Recent attacks that exploit representation editing, activation subtraction, or multi-turn adaptive optimization are not included, making it difficult to assess LatentBreak’s true competitiveness.

**Questions:**

While I appreciate the thorough evaluation and clear writing, the lack of new conceptual contribution and relatively outdated baselines reduce the paper’s impact.

---

> ### Author Response · Authors · 2025-11-21
> **Response to Reviewer oY46**
>
> We thank the reviewer for their feedback. In the following response, we address each of the listed weaknesses.
>
> **W1-W2**. *Limited novelty beyond representation shifting; relation to prior work*:  We respectfully disagree with the characterization of our contribution as incremental with respect to prior work, such as Arditi et al. or Li et al. These studies analyze refusal behavior and identify latent directions associated with safety mechanisms, showing how manipulating internal representations can suppress or induce refusal. In contrast, LatentBreak does not intervene on model activations: it manipulates the input based on latent-space feedback, operationalizing representational insights into an iterative substitution algorithm that generates natural, low-perplexity jailbreak prompts.
> This distinction places our contribution in a different category: LatentBreak builds an input-level attack mechanism rather than a representation-manipulation or model steering technique.
>
> Hence, crucially, its novelty should be evaluated against jailbreak methods that operate through input manipulation, rather than against steering works, whose primary goal is to analyze or modify internal representations.
>
> We also remark that, although interpretability is not the primary focus of our work, we still provide an initial mechanistic analysis in **Appendix H**. There, we compare the internal representations induced by GCG, SAA, AutoDAN, and LatentBreak, showing that typical jailbreaks drive harmful prompts into tight clusters near the harmless region, whereas LatentBreak does not exhibit this clustering effect. Instead, the representations corresponding to successful LatentBreak attacks tend to remain comparatively sparse and dispersed, highlighting a qualitatively different latent behavior relative to existing jailbreak methods.
>
> **W3**. *Baseline Selection is dated*: We still find it necessary to emphasize that LatentBreak is an input-manipulation attack, and thus its competitiveness should be assessed against other input-level jailbreak methods rather than representation-editing techniques, which belong to a different category of interventions. Within this scope, we selected the strongest and most widely adopted attacks according to established benchmarks. In particular, SAA (ICLR 2025) was included precisely because it is both recent and among the most effective attacks reported in the literature, as also noted in [ext_ref_2]. The remaining baselines were chosen to form a heterogeneous pool of high-performing input manipulation methods, consistent with the comparative analyses presented in [ext_ref_1].
>
> **Final Remarks**: We thank the reviewer for the comments. We believe our clarifications address the raised points, and we hope they will help the reviewer reconsider their evaluation.
>
> ## References
>
> [ext_ref_1]  Mazeika, M., Phan, L., Yin, X., Zou, A., Wang, Z., Mu, N., ... & Hendrycks, D. (2024). Harmbench: A standardized evaluation framework for automated red teaming and robust refusal. arXiv preprint arXiv:2402.04249.
>
> [ext_ref_2] Chu, J., Liu, Y., Yang, Z., Shen, X., Backes, M., & Zhang, Y. (2025, July). JailbreakRadar: Comprehensive assessment of jailbreak attacks against LLMs. In Proceedings of the 63rd Annual Meeting of the Association for Computational Linguistics (Volume 1: Long Papers) (pp. 21538-21566).

---

### Official Review · Reviewer_7jw3 · 2025-11-01

**Soundness:** 3
**Presentation:** 3
**Contribution:** 2
**Rating:** 2
**Confidence:** 4

**Summary:**

This paper introduces LatentBreak (LatB), a white-box jailbreaking method designed to bypass the perplexity defense. This proposed method modifies the original harmful prompt using token-level substitution guided by Latent-Space Feedback.

**Strengths:**

The proposed method is efficient and achieves higher ASR over all baseline attacks against perplexity detection.

**Weaknesses:**

1. Strict White-Box Dependency: LatB is strictly a white-box attack, requiring full access to the target model's internal intermediate activations for distance calculation. This severely limits its practical applicability against most commercially deployed LLMs (e.g., closed APIs) and is a significant constraint for real-world threat modeling.
2. Incomplete Baseline Comparison: The paper primarily focuses on comparing against high-perplexity automated attacks (GCG, SAA, AutoDAN) and a few common black-box approaches (PAIR, TAP). It lacks comparison with modern black-box semantic or persuasion-based jailbreaks, such as "How johnny can persuade LLMs to jailbreak them: Rethinking persuasion to challenge AI safety by humanizing LLMs." These methods also generate low-perplexity, and natural prompts, potentially representing a more direct threat and more relevant baseline for an attack focused on generating natural-sounding prompts. The absence of this comparison diminishes the demonstrated novelty in the "naturalness" dimension of the attack.

**Questions:**

1. The authors mention the latent space of a specific layer $l$ is used, but the justification for the choice of $l$ (e.g., the 28th layer of a 32-layer model) is not thoroughly discussed. Did the authors perform a sensitivity analysis on the layer index $l$?

2. The centroid $\mu$ is crucial. How sensitive is the attack's success rate to the size and content of the $\mathcal{D}_{harmless}$ dataset used to compute $\mu$?

---

> ### Author Response · Authors · 2025-11-21
> **Response to Reviewer 7jw3**
>
> We thank the reviewer for their feedback. In the following response, we address each of the listed weaknesses and answer all the questions raised.
>
> **W1** - *White-box dependency*: The reviewer suggests that the white-box threat model used by LatentBreak is a severe limitation. We respectfully disagree and thus find it necessary to remark why it is important to test models against white-box adversaries.
>
> It is widely recognized that meaningful robustness evaluation requires testing against attackers with maximal capabilities. This principle is central in adversarial machine learning research, where white-box attacks are used to establish empirical upper bounds on robustness. Relying solely on black-box attacks risks overestimating a model’s safety, as also emphasized in evaluations of adversarial robustness for ML models [ext_ref_1, ext_ref_2]. Recent LLM security work has explicitly adopted this philosophy, demonstrating that white-box adaptive adversaries reveal vulnerabilities that black-box evaluations fail to uncover [ext_ref_3]. Also, importantly, it is fundamental to test models against such worst-case attacks to develop, in response, safety mechanisms that effectively generalize.
>
> Finally, we strongly believe that research should not be confined to attacking commercial models. Improving our understanding of their failure modes in open source directly benefits the broader LLM community, including the safety of closed-source models.
>
> **W2** - *Incomplete baselines*: We would like to clarify an important point regarding the baseline comparison. AutoDAN is explicitly designed as a low-perplexity jailbreak, and SAA has also been shown, in [ext_ref_6], to produce low average perplexity due to its verbose template. Notably, our results show that both of these attacks, despite their lower perplexity, remain detectable under our windowed perplexity filter. Hence, the reviewer’s comment specifying that “the paper primarily focuses on comparing against high-perplexity automated attacks (GCG, SAA, AutoDAN)” is inaccurate for two out of the three mentioned attacks.
>
> Regarding black-box attacks, the reviewer suggests using “modern” approaches, such as PAP [ext_ref_4]. While appreciating the reference, we specify that (i) PAIR and TAP have both been shown to be substantially more powerful than the referenced PAP attack in [ext_ref_5] and (ii) that both PAIR and TAP already generate natural and low-perplexity prompts. Hence, this justifies our attack choice by a wide margin and, importantly, does not support the reviewer’s suggestion of using PAP to “represent a more direct and relevant threat”.
>
> **Q1**: As shown in Appendix D, we select the layer based on the resulting ASR and the distance between the harmless and harmful prompts’ distributions, thus targeting better separability between the two distributions.
>
> **Q2**: Thanks to the reviewer for raising this interesting question. In internal experiments, to our surprise, we noticed that the size of the dataset had little to no impact on the overall centroid computation. We will add that in the final version of the paper.
>
> **Final Remarks**: We thank the reviewer for the comments. We believe our clarifications address the raised points, and we hope they will help the reviewer reconsider their evaluation.
>
> ## References
>
> [ext_ref_1]: Biggio, B., & Roli, F. (2018, October). Wild patterns: Ten years after the rise of adversarial machine learning. In Proceedings of the 2018 ACM SIGSAC Conference on Computer and Communications Security (pp. 2154-2156).
>
> [ext_ref_2]: Carlini, N., Athalye, A., Papernot, N., Brendel, W., Rauber, J., Tsipras, D., ... & Kurakin, A. (2019). On evaluating adversarial robustness. arXiv preprint arXiv:1902.06705.
>
> [ext_ref_3]: Nasr, M., Carlini, N., Sitawarin, C., Schulhoff, S. V., Hayes, J., Ilie, M., ... & Tramèr, F. (2025). The attacker moves second: Stronger adaptive attacks bypass defenses against llm jailbreaks and prompt injections. arXiv preprint arXiv:2510.09023.
>
> [ext_ref_4]:  Zeng, Y., Lin, H., Zhang, J., Yang, D., Jia, R., & Shi, W. (2024, August). How johnny can persuade llms to jailbreak them: Rethinking persuasion to challenge ai safety by humanizing llms. In Proceedings of the 62nd Annual Meeting of the Association for Computational Linguistics (Volume 1: Long Papers) (pp. 14322-14350).
>
> [ext_ref_5] Mazeika, M., Phan, L., Yin, X., Zou, A., Wang, Z., Mu, N., ... & Hendrycks, D. (2024). Harmbench: A standardized evaluation framework for automated red teaming and robust refusal. arXiv preprint arXiv:2402.04249.
>
> [ext_ref_6]: Chao, P., Debenedetti, E., Robey, A., Andriushchenko, M., Croce, F., Sehwag, V., ... & Wong, E. (2024). Jailbreakbench: An open robustness benchmark for jailbreaking large language models. Advances in Neural Information Processing Systems, 37, 55005-55029.

---

### Official Review · Reviewer_ByYu · 2025-11-01

**Soundness:** 2
**Presentation:** 3
**Contribution:** 2
**Rating:** 4
**Confidence:** 4

**Summary:**

This paper introduces LatentBreak, a novel white-box jailbreaking attack designed to bypass the safety alignment of Large Language Models (LLMs). The authors first demonstrate that existing state-of-the-art attacks, which typically rely on optimizing adversarial suffixes or using long templates, are easily detected by robust perplexity-based filters due to the unnatural, high-perplexity tokens they introduce. To overcome this, LatentBreak operates by making strategic, semantically-equivalent word substitutions within the original harmful prompt. The core innovation lies in its feedback mechanism: instead of optimizing based on output logits, LatentBreak is guided by feedback from the model's latent space, iteratively selecting word replacements that shift the prompt's internal representation closer to a pre-computed centroid of "harmless" prompts. The experimental results show that this method produces shorter, more natural, low-perplexity jailbreak prompts that are highly effective at evading perplexity-based defenses and achieve a higher attack success rate than other methods under these defensive conditions.

**Strengths:**

Rather than chasing output logits or refusal tokens, this paper steers internal representations toward “harmless-like” regions. And this paper also introduces MaxPPL_W and shows it largely neutralizes suffix-style attacks (often ~0.0% ASR), yet the proposed attack retains high ASR (e.g., 71.1% on Mistral-7B, 66.7% on Vicuna-13B).

**Weaknesses:**

1. The Perplexity Defense as an Impractical "Strawman". The work frames itself as a necessary evolution to bypass perplexity-based filtering, but it's questionable whether such filtering is a realistic defense in the first place. Model service providers generally aim to maximize utility, allowing users to input anything from complex source code to creative prose. In fact, human-written text often has higher perplexity than AI-generated text [1], so high perplexity alone is not evidence that a user’s input is illegal. Implementing a strict PPL filter would be commercially and practically unviable, as it would block a large number of legitimate users. This makes LatentBreak's primary contribution (i.e., bypassing this strict PPL filter) feel like a solution to a non-problem.

2. Questionable Contribution Given the White-Box/Defense Trade-off. The attack is a pure white-box method, which inherently limits its practical threat and makes it highly susceptible to "overfitting" on the specific latent states of a single, local model. This sacrifice in practical applicability (i.e., giving up the black-box setting) is justified only by the paper's central argument: that it is needed to bypass robust PPL defenses. However, as argued in the previous point, this defense is itself impractical. Therefore, the paper's sacrificing practical, black-box universality in order to defeat a non-viable strawman defense calls the significance of the overall contribution into question.

[1] Mitrović, Sandra, Davide Andreoletti, and Omran Ayoub. "Chatgpt or human? detect and explain. explaining decisions of machine learning model for detecting short chatgpt-generated text." arXiv preprint arXiv:2301.13852 (2023).

**Questions:**

1. The authors mention transferability as future work, but this seems like a critical missing piece. Since the attack optimizes against the specific latent space of a single victim model, how transferable are the resulting "natural" prompts? Does a prompt generated for Llama-3-8B have any success against Llama-3-70B, or against a black-box model like GPT-4?

2. The attack algorithm appears computationally exorbitant. It involves up to $I=30$ iterations, and in each iteration, it loops through all $N$ words, generating $K=20$ candidates. Each candidate requires a forward pass to get the latent representation and, critically, a call to an LLM judge ($\mathcal{J}_{intent}$). This seems orders of magnitude more expensive than GCG, which is purely gradient-based. Could the authors provide an analysis of the wall-clock time and computational cost (e.g., in GPU hours and API calls) to generate a single jailbreak, and compare this to the cost of SOTA attacks?

---

> ### Author Response · Authors · 2025-11-21
> **Response to Reviewer ByYu (1/2)**
>
> We thank the reviewer for their feedback. In the following responses, we address each of the listed weaknesses (1/2) and answer all the questions raised (2/2).
>
> **W1** - *PPL Filters impracticability*: We appreciate the reviewer’s concern. We agree that commercial model providers may not deploy strict perplexity filtering, as it could interfere with general usability. However, PPL filters have been widely discussed as scientific solutions for detecting suffix-based jailbreaks [ext_ref_1, ext_ref_2, ext_ref_3]. Dismissing perplexity filtering as “impractical” would, by the same logic, imply that a large body of prior work investigating both attacks and defenses not applicable to commercial models addresses a “non-problem”. We believe this is not the reviewer’s intention.
>
> In this regard, we also strongly believe that it is highly relevant to demonstrate that standard perplexity-based defenses are surprisingly effective against leading white-box jailbreaks, despite being conceptually simple.
> For instance, SAA jailbreaks utilize a verbose template with a low average perplexity compared to other suffix-based attacks. Due to this, previous perplexity-based defenses [ext_ref_3], which relied on the average perplexity as a filtering strategy, were unable to detect it (see Appendix E). With the MaxPPL_W filter, however, we introduce a perplexity-based filter that detects local spikes in perplexity throughout the input prompt, effectively detecting the SAA’s adversarial suffix appended at the end of the jailbreaks. Finally, we want to specify that although this filter is capable of detecting such attacks, we always reported its performance at a fixed 0.5% FPR, without affecting the utility of the defense (i.e., only a few harmless prompts are flagged as jailbreaks).
>
> **W2** - *Contributions of white-box attacks*:  We respectfully disagree with the reviewer’s comment about “white-box attacks being a questionable contribution”.
>
> It is widely recognized that meaningful robustness evaluation requires testing against attackers with maximal capabilities. This principle is central in adversarial machine learning research, where white-box attacks are used to establish empirical upper bounds on robustness. Relying solely on black-box attacks risks overestimating a model’s safety, as also emphasized in various seminal papers on adversarial robustness [ext_ref_4, ext_ref_5]. Recent LLM security work has explicitly adopted this philosophy, demonstrating that white-box adaptive adversaries reveal vulnerabilities that black-box evaluations fail to uncover [ext_ref_6]. Also, importantly, it is fundamental to test models against such worst-case attacks to develop, in response, safety mechanisms that effectively generalize.
>
> As a result, we strongly disagree with the reviewer’s comment about “sacrificing black-box setting only to support the paper’s central argument”.
>
>
> ## References
>
> [ext_ref_1]: Alon, Gabriel, and Michael Kamfonas. "Detecting language model attacks with perplexity." arXiv preprint arXiv:2308.14132 (2023).
>
> [ext_ref_2]: Liu, X., Xu, N., Chen, M., & Xiao, C. (2023). Autodan: Generating stealthy jailbreak prompts on aligned large language models. arXiv preprint arXiv:2310.04451.
>
> [ext_ref_3]: Jain, N., Schwarzschild, A., Wen, Y., Somepalli, G., Kirchenbauer, J., Chiang, P. Y., ... & Goldstein, T. (2023). Baseline defenses for adversarial attacks against aligned language models. arXiv preprint arXiv:2309.00614.
>
> [ext_ref_4]: Biggio, B., & Roli, F. (2018, October). Wild patterns: Ten years after the rise of adversarial machine learning. In Proceedings of the 2018 ACM SIGSAC Conference on Computer and Communications Security (pp. 2154-2156).
>
> [ext_ref_5]: Carlini, N., Athalye, A., Papernot, N., Brendel, W., Rauber, J., Tsipras, D., ... & Kurakin, A. (2019). On evaluating adversarial robustness. arXiv preprint arXiv:1902.06705.
>
> [ext_ref_6]: Nasr, M., Carlini, N., Sitawarin, C., Schulhoff, S. V., Hayes, J., Ilie, M., ... & Tramèr, F. (2025). The attacker moves second: Stronger adaptive attacks bypass defenses against llm jailbreaks and prompt injections. arXiv preprint arXiv:2510.09023.

---

> ### Author Response · Authors · 2025-11-21
> **Response to Reviewer ByYu (2/2)**
>
> **Q1**: We did not primarily investigate transferability because the main contribution of our work is the introduction of a new white-box attack that leverages the latent space of a specific victim model. Explicitly targeting transferability would require designing and evaluating multi-surrogate optimization procedures, which represent a substantially larger and different experimental effort. Nevertheless, this line of inquiry is valuable, and we explicitly highlight it as future work.
> That said, we conducted preliminary transferability tests of LatentBreak by taking the jailbreak prompts generated for different open-source models and evaluating them against two closed-source models: Claude 3.7 Sonnet and Gemini-1.5-Pro. Below, we report the transfer ASR per surrogate and the intersection ASR (i.e., the ASR obtained by combining the intersection of prompts from all surrogate models).
>
> **Claude 3.7 Sonnet**
>
> | Surrogate model | ASR (%) |
> |-----------------|---------|
> | Gemma-7B        | 0.645   |
> | Llama-2         | 0.645   |
> | Llama-3         | 6.452   |
> | Vicuna-13B      | 0.000   |
> | Qwen-7B         | 0.645   |
> | Mistral-7B      | 1.290   |
> | Phi-mini        | 1.290   |
> | **Intersection**| **9.032** |
>
> **Gemini-1.5-Pro**
> | Surrogate model | ASR (%) |
> |-----------------|---------|
> | Gemma-7B        | 13.548  |
> | Llama-2         | 1.935   |
> | Llama-3         | 10.968  |
> | Vicuna-13B      | 6.452   |
> | Qwen-7B         | 8.387   |
> | Mistral-7B      | 9.677   |
> | Phi-mini        | 11.613  |
> | **Intersection**| **29.677** |
>
> These results suggest that LatentBreak’s prompts can transfer to closed-source models to a non-negligible degree, even though the attack is not explicitly optimized for transferability. Hence, we find the above result relevant. We expect that transfer performance could be substantially improved by incorporating a larger and more diverse pool of surrogate models, or by selecting surrogates specifically from model families that show stronger alignment with the target closed-source system.
>
> **Q2**:  LatentBreak **does not** call the evaluation judge $\mathcal{J_{judge}}$ each candidate iteration $K$, nor does it for each input prompt word $N$. It simply calls the evaluation judge in each of the 30 iterations, which is substantially different. What is called in each of the $K$ iterations, instead, is the intent judge $\mathcal{J}_{intent}$, which has a lower cost.
>
> In turn, LatentBreak is orders of magnitude faster than GCG, as we show in the following table:
> | Attack | Avg. time per prompt |
> |--------|------------------------|
> | LatB   | 2.57 minutes           |
> | GCG    | 36.2 minutes           |
>
> From this analysis, performed on Vicuna-13B with a subset of samples from AdvBench, GCG takes approximately 4.25 seconds per iteration, a value consistent with the one reported in [ext_ref_7], which measures an average iteration time of 4.8 seconds, on a model from the same family.
>
> **Final Remarks**: We thank the reviewer for the comments. Several of the main criticisms, however, appear to stem from a different interpretation of the motivation and scope of our work, particularly regarding the role of perplexity defenses and the value of white-box evaluation. We believe that this difference in perspective led to an assessment that does not fully capture the contribution or its scientific relevance. We hope our clarifications help align these perspectives and induce a reconsideration of the evaluation.
>
> ## References
> [ext_ref_7]: Zhao, Y., Zheng, W., Cai, T., Long, X., Kawaguchi, K., Goyal, A., & Shieh, M. Q. (2024). Accelerating greedy coordinate gradient and general prompt optimization via probe sampling. Advances in Neural Information Processing Systems, 37, 53710-53731.

---

> > ### Comment · Reviewer_ByYu · 2025-11-27
> >
> > Dear Authors,
> >
> > Thank you for your detailed rebuttal. It addresses my concerns, and I am satisfied with the clarifications provided.
> >
> > Thank you, well done.

---

### Official Review · Reviewer_KRMS · 2025-11-01

**Soundness:** 3
**Presentation:** 3
**Contribution:** 2
**Rating:** 2
**Confidence:** 3

**Summary:**

In this paper, the authors propose LatentBreak, a white-box jailbreak attack to decrease the perplexity of the crafted prompt. In detail, the authors propose to substitute words with the help of the substitution models, maintaining their original meaning and decreasing the harmfulness of the crafted input.

**Strengths:**

1 The soundness of the proposed method is good.

2 The writing is easy to follow.

3 The experiments are quite solid.

**Weaknesses:**

1 The accessibility of model representation limits the application scenario of this attack. It can only be applied to attack open-source models. However, as far as I know, there are a lot of approaches that can bypass the internal alignment capability of the open-source models. This limitation will make this attack less attractive to the readers.

2 I think the perplexity-based filtering method proposed in this paper is not a realistic defense. From Table 2, we can summarize that the ASR after defense has negative correlations with the response length. It reveals a prominent disadvantage of this method: It brings a high false positive rate when the LLMs require a long thinking process to answer complex requests like an International Mathematical Olympiad Problem.

3 Insufficient discussions with the most related works. In [2], they also propose a jailbreak attack from a representation space and more discussions are needed to highlight the novelty of this paper.

4 Actually, there are lots of existing works that can not only keep the length of the suffix short and achieve low PPL. For example, AutoDAN [1] combines the dual goals of jailbreak and readability and achieves a low-perplexity prompt. The in-context attack in [2] adds adversarial context to the history of the models without affecting the PPL of the adversarial prompt.


[1] AutoDAN: Interpretable Gradient-Based Adversarial Attacks on Large Language Models

[2] xJailbreak: Representation Space Guided Reinforcement Learning for Interpretable LLM Jailbreaking

[3] Jailbreak and guard aligned language models with only few in-context demonstrations

**Questions:**

1 In this paper, since the authors propose to substitute the sensitive words with semantically equivalent ones, can the attacks resist the prompt-based defense, such as PAT and RPO?

2 Noticing that the algorithm only incorporates substitution to renew the prompt. But as far as I know, the initialization of the prompt is significant in the optimization process. Can you explain more details?

---

> ### Author Response · Authors · 2025-11-21
> **Response to Reviewer KRMS (1/2)**
>
> We thank the reviewer for their feedback. In the following responses, we address each of the listed weaknesses (1/2) and answer all the questions raised (2/2).
>
> **W1** - *White-box jailbreaks and applicability*: The reviewer highlights that the LatentBreak requirement of having access to the model’s internals limits the applicability and attractiveness of the attack. **We respectfully argue that this setting does not reduce the significance or applicability of our findings.**
>
> It is widely recognized that meaningful robustness evaluation requires testing against attackers with maximal capabilities. This principle is central in adversarial machine learning research, where white-box attacks are used to establish empirical upper bounds on robustness. Relying solely on black-box attacks risks overestimating a model’s safety, as also emphasized in seminal papers on the evaluation of adversarial robustness for ML models (e.g., [ext_ref_1, ext_ref_2]). Recent LLM security work has explicitly adopted this philosophy, demonstrating that white-box adaptive adversaries reveal vulnerabilities that black-box evaluations fail to uncover [ext_ref_3]. Also, importantly, it is fundamental to test models against such worst-case attacks to develop, in response, safety mechanisms that effectively generalize.
>
> Finally, we strongly believe that research should not be confined to attacking commercial models. Improving our understanding of their failure modes in open source directly benefits the broader LLM community, including the safety of closed-source models.
>
> **W2** - *Perplexity-based filtering*: The reviewer argues that the considered perplexity-based filtering is not realistic and may lead to a high false positive rate when the LLM generates complex **output** responses. However, we need to clarify an important point: our perplexity-based filter is designed to filter only **input prompts** (not model output responses). In Table 2 (as well as in Table 1), we report the ASR achieved by an attack and the prompt size increase of the **input** jailbreak. Hence, no negative correlation can be inferred between ASR and response length from Table 2.
>
> In turn, the reviewer's concern about the false positive rate (FPR) is not relevant to our case.
>
> **W3/4** - *Insufficient discussion of related work*:
> We thank the reviewer for pointing out the xJailbreak attack reference [2]. We were not aware of this attack, and we will integrate this work into the related work section of our submission.
> Regarding AutoDAN-Zhu [1], we specify that the AutoDAN version we considered (i.e., AutoDAN-Liu) represents a more established attack and has consistently been shown to outperform the Zhu version in terms of both ASR and average perplexity (e.g., in [ext_ref_4]). This justifies our decision to consider only the AutoDAN-Liu attack.
>
> Finally, while the ICA attack [3] claims to achieve lower perplexity values compared to suffix-based jailbreaks, this claim is, to our surprise, not empirically supported in the paper, thereby limiting the relevance of the comparison with ICA.
>
> ## References
>
> [ext_ref_1]: Biggio, B., & Roli, F. (2018, October). Wild patterns: Ten years after the rise of adversarial machine learning. In Proceedings of the 2018 ACM SIGSAC Conference on Computer and Communications Security (pp. 2154-2156).
>
> [ext_ref_2]: Carlini, N., Athalye, A., Papernot, N., Brendel, W., Rauber, J., Tsipras, D., ... & Kurakin, A. (2019). On evaluating adversarial robustness. arXiv preprint arXiv:1902.06705.
>
> [ext_ref_3]: Nasr, M., Carlini, N., Sitawarin, C., Schulhoff, S. V., Hayes, J., Ilie, M., ... & Tramèr, F. (2025). The attacker moves second: Stronger adaptive attacks bypass defenses against llm jailbreaks and prompt injections. arXiv preprint arXiv:2510.09023.
>
> [ext_ref_4] Guo, X., Yu, F., Zhang, H., Qin, L., & Hu, B. (2024). Cold-attack: Jailbreaking llms with stealthiness and controllability. arXiv preprint arXiv:2402.08679.

---

> ### Author Response · Authors · 2025-11-21
> **Response to Reviewer KRMS (2/2)**
>
> **Q1**: We ran LatentBreak against PAT with Vicuna-13B and Qwen-7B as victim models. We employed the same defense prefix provided by the PAT’s official implementation for transferable settings. We ran the evaluation on a subset of samples from AdvBench.
> We report LatentBreak’s ASR with and without PAT in the following Table:
>
> | Victim Model        | ASR (%) |
> |---------------------|---------|
> | Vicuna-13B          | 72.0    |
> | Vicuna-13B + PAT    | 26.0    |
> | Qwen-7B             | 80.0    |
> | Qwen-7B + PAT       | 48.0    |
>
> Through this result, we want to highlight how LatentBreak maintains a non-trivial attack success rate even in the presence of PAT, indicating that LatB can remain resilient against prompt-based defenses that rely on fixed defensive prefixes. This suggests that substituting sensitive words with semantically equivalent alternatives allows LatentBreak to partially circumvent such mechanisms. We will add these results to a specific Appendix section.
>
> **Q2**: We remark that the initialization of the LatentBreak attack is the representation of the original prompt itself. From that, LatB’s algorithm selects the replacements that mostly decrease the distance from the initial prompt’s representation to the centroid of harmless representations.
>
> **Final Remarks**: We thank the reviewer for their comments. Several concerns, however, stem from misunderstandings of our setting. In particular, the comments regarding the role of white-box evaluation and of our defense analysis led to an incomplete assessment of the contribution. We believe that our clarifications address these misconceptions and more accurately reflect the intent and relevance of the work. We hope this helps the reviewer reconsider their evaluation.

---

> > ### Comment · Reviewer_KRMS · 2025-11-27
> > **Thank you for your rebuttal**
> >
> > Dear authors,
> >
> > Thank you for your rebuttal. Since there is still time to incorporate the reviewers’ suggestions, I would recommend submitting a revised version of the paper.
> >
> > Best,
> >
> > Reviewer KRMS

---

> > > ### Author Response · Authors · 2025-11-27
> > >
> > > Dear Reviewer,
> > >
> > > Thank you. A revised version incorporating the reviewers’ feedback has been submitted.

---

### Author Response · Authors · 2025-11-27
**Revisions overview**

We thank the reviewers for their valuable feedback. Following the suggestions provided during the review process, we have updated the related work section and the appendix to include the following additions:
- Appendix I: an evaluation of LatentBreak against the Prompt Adversarial Tuning defense.
- Appendix J: a preliminary transferability analysis on closed-source models.
- Appendix K: a runtime cost comparison between LatentBreak and GCG.

We hope these additions address the reviewers’ requests and further clarify the contributions of our work.

---

### Note · Authors · 2026-01-15

I have read and agree with the venue's withdrawal policy on behalf of myself and my co-authors.